# Untranslated Regions of a Segmented Kindia Tick Virus Genome Are Highly Conserved and Contain Multiple Regulatory Elements for Viral Replication

**DOI:** 10.3390/microorganisms12020239

**Published:** 2024-01-23

**Authors:** Anastasia A. Tsishevskaya, Daria A. Alkhireenko, Roman B. Bayandin, Mikhail Yu. Kartashov, Vladimir A. Ternovoi, Anastasia V. Gladysheva

**Affiliations:** 1State Research Center of Virology and Biotechnology «Vector», 630559 Kol’tsovo, Russia; tsishevskaya_aa@vector.nsc.ru (A.A.T.); alhireenko_da@vector.nsc.ru (D.A.A.); bayandin_rb@vector.nsc.ru (R.B.B.); kartashov_myu@vector.nsc.ru (M.Y.K.); tern@vector.nsc.ru (V.A.T.); 2Physics Department, Novosibirsk State University, 630090 Novosibirsk, Russia; 3Natural Sciences Department, Novosibirsk State University, 630090 Novosibirsk, Russia

**Keywords:** ixodid ticks, Flaviviridae, Jingmenvirus group, Kindia tick virus, orthoflavivirus, flavi-like virus, segmented virus, phylogenetics, RNA structure, untranslated region

## Abstract

Novel segmented tick-borne RNA viruses belonging to the group of Jingmenviruses (JMVs) are widespread across Africa, Asia, Europe, and America. In this work, we obtained whole-genome sequences of two Kindia tick virus (KITV) isolates and performed modeling and the functional annotation of the secondary structure of 5′ and 3′ UTRs from JMV and KITV viruses. UTRs of various KITV segments are characterized by the following points: (1) the polyadenylated 3′ UTR; (2) 5′ DAR and 3′ DAR motifs; (3) a highly conserved 5′-CACAG-3′ pentanucleotide; (4) a binding site of the La protein; (5) multiple UAG sites providing interactions with the MSI1 protein; (6) three homologous sequences in the 5′ UTR and 3′ UTR of segment 2; (7) the segment 2 3′ UTR of a KITV/2017/1 isolate, which comprises two consecutive 40 nucleotide repeats forming a Y-3 structure; (8) a 35-nucleotide deletion in the second repeat of the segment 2 3′ UTR of KITV/2018/1 and KITV/2018/2 isolates, leading to a modification of the Y-3 structure; (9) two pseudoknots in the segment 2 3′ UTR; (10) the 5′ UTR and 3′ UTR being represented by patterns of conserved motifs; (11) the 5′-CAAGUG-3′ sequence occurring in early UTR hairpins. Thus, we identified regulatory elements in the UTRs of KITV, which are characteristic of orthoflaviviruses. This suggests that they hold functional significance for the replication of JMVs and the evolutionary similarity between orthoflaviviruses and segmented flavi-like viruses.

## 1. Introduction

Over the past five years, many novel Flaviviridae viruses possessing atypical genome structures have been discovered using high-throughput sequencing, which has challenged the traditional principles of virus classification [1]. The Flaviviridae family comprises epidemiologically significant orthoflaviviruses, such as West Nile virus, Zika virus, dengue virus, yellow fever virus, tick-borne encephalitis, etc. Jingmenviruses (JMVs) are novel RNA viruses that were first discovered in *Rhipicephalus microplus* ticks from the Jingmen region of Hubei Province in China in 2014 [2]. According to the International Committee on Taxonomy of Viruses (ICTV), the JMV group is considered to comprise unclassified viruses of the Flaviviridae family and currently includes Jingmen tick virus (JMTV), Mogiana tick virus (MGTV), Kindia tick virus (KITV), Alongshan virus (ALSV), Yanggou tick virus (YGTV), Takachi virus (TAKV), Harz mountain virus (HMV), Sichuan tick virus (SCTV), SCWL tick virus (SCWLTV), etc. The genetic material of these viruses has been found not only in ticks but also in mosquitoes, cattle, bats, rodents, and humans [3,4,5,6]. To date, JMVs have been discovered in Asia, Europe, Central and South America, and Africa. Recently, the discovery of JMVs in Russia and their potential pathogenicity to humans have been first reported [7,8,9]. JMVs are a new group of viruses that bear a risk of causing a pandemic. Because they are closely related to arthropods, they are found in animals in close contact with humans and cause febrile illness in patients.

JMVs are fundamentally different from “classical” orthoflaviviruses due to a segmented single-stranded RNA positive-sense genome. The genome consists of four (or five) segments. Each segment comprises one or more open reading frames (ORFs) and characteristic 5′ and 3′ untranslated regions (5′ and 3′ UTRs) and is presumably packaged into a separate viral particle [4,10]. The proteins encoded by segments 1 and 3 are genetically and functionally related to nonstructural NS3 and NS5 proteins of classical orthoflaviviruses from the Flaviviridae family. The RNA-dependent RNA polymerase gene was shown to be integrated into the genome of *Ixodes ricinus* ticks. The other two segments, encoding structural VP1–VP3 proteins, are of unknown origin [11]. Very little information is currently available on the secondary structure of the JMV 5′ and 3′ UTRs that function as cis-acting elements during viral genome replication, translation, and viral life cycle regulation [6,12]. Knowledge of the structure of JMV 5′ and 3′ UTRs will help us to study the endemic potential of JMVs and to develop antiviral drugs and vaccines.

Kindia tick virus (KITV) is a novel unclassified tick-borne flavi-like virus from the JMV group of the Flaviviridae family. It was first discovered in ixodid ticks *Rhipicephalus geigyi* collected from cattle around the city of Kindia, Republic of Guinea, in 2017 [13]. Then, KITV genetic material was found in six more *Rhipicephalus geigyi* ticks from the 2021 collection [14]. KITV has a genome structure typical of that in JMVs; NS3 and NS5 proteins are structurally and functionally similar to those in orthoflaviviruses, which confirm their possible evolutionary relationship and taxonomic unity; and structural VP1–VP3 proteins of KITV have no analogues among known viral proteins [15]. In this study, the secondary structure of the KITV RNA 5′ and 3′ UTRs was for the first time modeled and analyzed, regulatory elements in the KITV 5′ and 3′ UTRs characteristic of orthoflaviviruses were discovered, and conserved motifs were identified in JMVs, including KITV.

## 2. Materials and Methods

### 2.1. Tick Collection

The collection of ixodid ticks was formed in 2018 from freshly slaughtered cattle in a slaughterhouse in Kindia, the Republic of Guinea. Sampled ticks were classified into species based on their morphological characteristics. Next, they were homogenized and stored at −80 °C until analyses were undertaken [16]. Tick collection was kindly provided by the Research Institute of Applied Biology of the Republic of Guinea.

### 2.2. Whole-Genome Sequencing

Total RNA was isolated from tick (*Rhipicephalus* spp.) homogenates using phenol–chloroform extraction and an ExtractRNA reagent (Evrogen, Moscow, Russia). Reverse transcription was performed using a MMLV RT kit (Evrogen, Moscow, Russia). Screening for the genetic material of segmented flavi-like viruses was carried out through a PCR using JMV_f—TGGACCAGGGCMGTIGGRGAGTA and JMV_r—GAAAACCTGRTAGTYIGGGTCGCA oligonucleotides [2]. Kindia tick virus cDNA fragments were amplified using a Q5 High-Fidelity DNA Polymerase kit (NEB, Ipswich, MA, UK), and original oligonucleotides were calculated by the authors. The data are shown in Appendix A. Amplicons were analyzed through electrophoresis in 2% agarose gel and purified using a Cleanup Standard kit (Evrogen, Moscow, Russia). The sequencing reaction was performed using a BigDye Terminator v3.1 kit (Thermo FS, Waltham, MA, USA). After the sequencing reaction, fragments were purified through direct reprecipitation with ethanol. Whole-genome sequencing was performed using an ABI 3500/3500xl device (Applied Biosystems, Waltham, MA, USA). Whole-genome sequence assembly and sequence chromatogram processing were performed using Lasergene 10 SeqMan (DNASTAR, Madison, WI, USA) and UGENE version: 1.31.1 (UNIPRO, Moscow, Russia) software. The search for open reading frames and translation into amino acid sequences were performed using Vector NTI (Invitrogen, Waltham, MA, USA). All whole-genome sequences were deposited in the GenBank database under accession numbers MW341206–MW341213.

### 2.3. Analysis of Whole-Genome Sequencing

All deposited whole-genome sequences of segmented flavi-like viruses were downloaded from the GenBank database, including Kindia tick virus, JMTV, MGTV, ALSV, YGTV, TAKV, HMV, SCTV, and SCWLTV. A total of 541 whole-genome sequences were downloaded as follows: 128 sequences for segment 1, 124 for segment 2, 144 for segment 3, and 145 for segment 4. Multiple alignment of whole-genome sequences and 5′–3′ UTRs was carried out in MEGA X (PSU, Philadelphia, PA, USA) using ClustalW. The number of selected objects of the segment 1 5′ and 3′ UTRs was as follows: 79 for JMTV, 5 for TAKV, 6 for YGTV, 10 for ALSV, 7 for HMV, 3 for MGTV, 18 for KITV. The number of selected objects of the segment 2 5′ and 3′ UTRs was as follows: 82 for JMTV, 3 for YGTV, 9 for ALSV, 5 for HMV, 3 for MGTV, 18 for KITV, 3 for SCTV, and 1 for SCWLTV. The number of selected objects of the segment 3 5′ and 3′ UTRs was as follows: 101 for JMTV, 3 for YGTV, 8 for ALSV, 7 for HMV, 3 for MGTV, 18 for KITV, 3 for SCTV, and 1 for SCWLTV. The number of selected objects of the segment 4 5′ and 3′ UTRs was as follows: 102 for JMTV, 3 for YGTV, 8 for ALSV, 7 for HMV, 3 for MGTV, 18 for KITV, 3 for STCV, and 1 for SCWLTV.

For the construction of a phylogenetic tree, complete nucleotide sequences of the segment 1 open reading frame from seventeen representative isolates of segmented flavi-like viruses were aligned and analyzed through the maximum likelihood method with 1000 bootstrap replicates using the MEGA X molecular evolutionary genetic analysis software (PSU, Philadelphia, PA, USA). The tree optimization algorithm and distance correction (G + R + I) were selected using the JMODELTEST software version: 2.1.10 (University of Vigo, Vigo, Spain). Isolates in the phylogenetic tree were labeled using a GenBank number followed by the isolate’s name, country, origin, and year of isolation.

### 2.4. 5′ UTR–3′ UTR Sequences and Structures

The primary criterion for the selection of untranslated regions was their size. Isolates lacking an untranslated region or that were shorter than 35 nucleotides, which indicated their under-sequencing, were excluded from the sample. The threshold was empirically chosen.

Novel, unresolved 5′–3′ UTR motifs (recurring, fixed-length patterns) were searched using a software package provided by the MEME Suite web server (version: 5.5.2) (https://meme-suite.org/meme, accessed on 28 December 2023). Prediction of 5′–3′ UTR structures and that of sequence alignment were performed using the LocARNA tool (http://rna.informatik.uni-freiburg.de/LocARNA/Input.jsp, accessed on 28 December 2023). The resulting structural alignments were visualized as RNA secondary structures with RNAplot from the ViennaRNA package (http://rna.tbi.univie.ac.at, accessed on 28 December 2023) using the “most informative sequence” and “annotate covariance of base pairs” modes and the RNApuzzler plotting layout algorithm.

Secondary structures of the KITV RNA 5′ and 3′ UTRs of segments 1 and 2 were predicted using three independent tools as follows: ViennaRNA Fold (http://rna.tbi.univie.ac.at, accessed on 28 December 2023), UNA MFOLD 3.6 (http://www.unafold.org/mfold, accessed on 28 December 2023), and RNAstructure (https://rna.urmc.rochester.edu, accessed on 28 December 2023) [17,18,19]. Simulation was performed at a folding temperature of 37 °C and under ionic conditions of 1 M NaCl without divalent ions. The parameters of “maximal distance between paired bases” (MDBPB) and “percent suboptimality” (%S) were selected manually. An MDBPB of 60 to 100 and a %S of up to 50% were established. The upper bound on the number of computed folds and the upper bound on the total number of single-stranded bases that are allowed in a bulge or interior loop were set at 25. Other parameters were set as default, and the initial free energy ∆G was set as the minimum value. For detailed analysis, secondary structures were linearized using the VARNA 3.9 software (http://varna.lri.fr, accessed on 28 December 2023) and redrawn using graphic editors [20].

## 3. Results

### 3.1. Phylogenetic Tree and Substitution Rate

In this study, we obtained whole-genome sequences of two new Kindia tick virus (KITV) isolates, KITV/2018/1 and KITV/2018/2, from homogenates of *Rhipicephalus* spp. ticks collected from cattle in Kindia, Republic of Guinea, in 2018. The total genome lengths for all the segments amounted to ~11,100 b.p., KITV/2018/1—11,173 b.p., and KITV/2018/2—11,129 b.p., which are consistent with the genome size of classical orthoflaviviruses.

Phylogenetic analysis of the open reading frames encoding different viral proteins allowed KITV to be taxonomically classified as a flavi-like virus of the JMV group of the Flaviviridae family. The result of the phylogenetic analysis is shown in Figure 1. The KITV/2018/1 and KITV/2018/2 isolates and a previously discovered KITV/2017/1 isolate form a separate genogroup (MK673133–MK673136). The homology levels between the KITV/2018/1 and KITV/2018/2 isolates and the KITV/2017/1 isolate were 96–99% for nucleotide sequences (different segments) and 97–99% for amino acid sequences. In this case, amino acid substitutions were found in all the viral proteins as follows: VP1 (9 substitutions and 11 substitutions), VP2 (1 and 1), VP3 (4 and 6), NS3 (7 and 9), and NS5 (3 and 11). There were also amino acid differences between the KITV/2018/1 and KITV/2018/2 isolates in the VP1 (2), VP3 (2), NS3 (4), and NS5 (5) proteins. The data are shown in Appendix A. The Brazilian JMTV_1 (MH155890-MH155893) and the MGTV/V4/11 (JX390986, KY523073, JX390985, KY523074) isolates are most closely related to KITV. The level of amino acid sequence homology of the KITV/2018/1 and KITV/2018/2 isolates with the Brazilian JMTV_1 and MGTV/V4/11 isolates was 90–95%.

### 3.2. KITV 5′ and 3′ UTR Sequences

Multiple alignment of KITV and MGTV 5′ and 3′ UTRs revealed that lengths of the 5′ and 3′ UTRs varied significantly due to various insertions, deletions, and nucleotide substitutions. The data are shown in Table 1. KITV is characterized by a short 5′ UTR (91–154 b.p.) and a longer 3′ UTR (up to 387 b.p.), which are consistent with the length of untranslated regions in Zika virus and tick-borne encephalitis virus. The data are shown in Table 1. The 3′ UTR of KITV and MGTV contains a ~20 b.p. polyA tail, which is not typical of viruses of the Flaviviridae family. Moreover, the 3′ UTR of all KITV segments was found to harbor multiple UAG sites (two to six) for interaction with the RNA-binding protein Musashi-1 (MSI1) associated with Zika virus neurotropism. The data are shown in Appendix A.

### 3.3. KITV Segment 1’s 5′ and 3′ UTR Structures

The secondary structure of segment 1’s 5′ UTR is represented by the two stem loops of SL-1 and SL-2. The model of the secondary structure is shown in Figure 2a. Hairpin 1 comprises a 5′-GUGC-3′ inverted sequence that comprises the La autoantigen binding site [21]. The CCAGG sequence is located 111 nucleotides downstream of the AUG start codon. In orthoflaviviruses, this sequence is called 5′ DAR. In the 3′ UTR, the complementary 5′-CCUGG-3′ sequence (3′ DAR) was found 50 nucleotides downstream of the stop codon. The 5′ DAR–3′ DAR regions are involved in flavivirus genome cyclization [22]. A distant location of the KITV 5′ DAR results in the emergence of three stem loops (SL-3, SL-4, SL-5) and a Y-structure. The 3′ UTR topology is conserved among the JMV isolates examined. This is represented by the Y-1 structure and SL-1 in all the KITV isolates and the MGTV/V4/11 isolate. The model of the secondary structure is shown in Figure 2b. A difference was found in the MGTV Yunnan2016 isolate in which this region was represented by two Y-structures. A highly conserved orthoflaviviral 5′-CACAG-3′ pentanucleotide was found in hairpin 3 of the Y-1 structure.

### 3.4. KITV Segment 2’s 5′ and 3′ UTR Structures

The secondary structure of segment 2’s 5′ UTR is represented by SL-1 and SL-2. The model of secondary structure is shown in Figure 3a. This segment comprises the La binding region at position 12 and downstream regions 5′ DAR—3′ DAR. The 5′ DAR is located 50 nucleotides downstream of AUG, which results in SL-4 and an extended free end. The AUG start codon occurs in SL-4, and the 5′ DAR is located in the free end. Similarly, the 3′ UTR comprises the 3′ DAR at position 2608 of the SL-6 nucleotide structure.

The secondary structure of the segment 2 3′ UTR is represented by the Y-1 and Y-2 structures in the MGTV Yunnan2016 isolate, the Y-1, Y-2, and Y-3 structures in the KITV/2017/1 and KITV/2018/2 isolates, and the SL-1, Y-1, and Y-2 structures in KITV/2018/1. Models of secondary structure are shown in Figure 3b–d. The Y-1 and Y-2 structures are conservative, and the Y-3 structure is variable. C47->U47 and C57->U57 substitutions in the 3′ UTR of KITV/2018/1 cause the replacement of Y-3 by SL-1. There is a modification in the Y-3 structure of the KITV/2018/1 and KITV/2018/2 isolates compared with KITV/2017/1. In addition, the orthoflaviviral 5′-CACAG-3′ pentanucleotide was found to occur in hairpin 7 of KITV/2017/1 and be absent in KITV/2018/1 and KITV/2018/2 due to a deletion in this region.

Four complementary and three homologous sequences (R1, R2, R3) were identified in the 5′ and 3′ UTRs. The discovered sequences are shown in Figure 3. The R1 sequence 5′-UGGCAAGUGC-3′ (10 b.p.) was found at nucleotide positions 6–15 in the 5′ UTR and positions 2749–2758 (2712–2721) in the 3′ UTR; the R2 sequence 5′-AAAGGAAAAAA-3′ (11 b.p.) was found at positions 82–92 and 2699–2709 (2662–2669), respectively; the R3 sequence 5′-AAAAAAGAACAAAAAAA-3′ (17 b.p.) was found at positions 101–117 and 2566–2576 in KITV/2017/1 and KITV/2018/1 with 88% homology. An A-to-U nucleotide substitution occurs in KITV/2018/2, which reduces homology to 82%. Moreover, there are two 40-nucleotide extended sequences in the 3′ UTR, which form the R4 repeat (positions 2450–2489 and 2487–2526). R4 is found only in KITV/2017/1 and leads to a modification of the Y-3 structure and the emergence of the orthoflaviviral 5′-CACAG-3′ pentanucleotide. In the KITV/2018/1 and KITV/2018/2 isolates, there is a 35-nucleotide deletion in the second R4 repeat region. This region in KITV/2017/1 involves hairpins 7 and 8 in Y-3 and causes the disappearance of 5′-CACAG-3′. The discovered region is shown in Figure 3b–d.

Two pseudoknots, PK1 and PK2, were found in the KITV segment 2’s 3′ UTR. PK1 is present in all the KITV isolates and occurs at positions 2416 (UAG) and 2630 (AUC) in KITV/2017/1 and at positions 2414 (UAG) and 2593 (AUC) in KITV/2018/1 and KITV/2018/2. The discovered regions are shown in Figure 3b–d. PK2 is found only in the KITV/2018/1 isolate, occurs at positions 2483 (GCA) and 2497 (UGC), and stabilizes the SL-1 structure. The discovered region is shown in Figure 3b.

### 3.5. KITV Segment 3’s 5′ UTR Structure

The secondary structure of segment 3’s 5′ UTR is represented by one stem loop and one Y-structure. The model of the secondary structure is shown in Figure 4. Hairpin 1 contains an inverted sequence of the La autoantigen binding site. The 5′-CCAGG-3′ sequence occurs 165 nucleotides downstream of the AUG start codon. No complementary 5′-CCTGG-3′ sequence (3′ DAR) was found in the 3′ UTR. The distant location of the 5′ DAR in KITV modifies the Y-structure through the emergence of four hairpins in it. The 5′-CACAG-3′ pentanucleotide was not found in the 3′ UTR, which may be due to the under-sequencing of this region. However, a similar sequence was found in the 5′ UTR of the KITV/2018/1 and KITV/2018/2 isolates (coordinates 12–16); in the KITV/2017/1 isolate, there was a substitution at position 14, leading to the disappearance of this pentanucleotide. The data are shown in Appendix A.

### 3.6. KITV Segment 4’s 5′ and 3′ UTR Structures

The secondary structure of segment 4’s 5′ UTR is represented by SL-1, SL-2, and SL-3, and that of segment 4’s 3′ UTR is represented by two Y-structures. Models of secondary structure are shown in Figure 5. The 5′ UTR SL-1 structure comprises the La binding region and the 5′-UGGCAAGUGC-3′ R1 sequence previously found in KITV segment 2’s 5′ and 3′ UTRs. The same R1 sequence is present in the second hairpin of the 3′ UTR Y-1 structure. No other regulatory elements were found.

### 3.7. Conserved Regions of KITV 5′ and 3′ UTRs

Based on the primary selection criteria, segment 1’s 5′ UTRs of the 25 JMTV and 2 ALSV isolates, segment 2’s 5′ UTRs of the 1 JMTV and 2 ALSV isolates, segment 3’s 5′ UTRs of the 27 JMTV isolates, segment 4’s 5′ UTRs of the 31 JMTV isolates, and segment 4’s 3′ UTRs of the 3 JMTV isolates were excluded from further analysis. Segment 4’s 3′ UTRs of six JMTV isolates were also excluded because an untranslated region was not identified due to a mutation in the VP3 protein.

#### 3.7.1. Segment 1’s 5′ UTR

Six conserved motifs were found in segment 1’s 5′ UTR. The discovered motifs are shown in Figure 6a. All analyzed KITV isolates were characterized by the presence of motifs 1, 2, and 4. The highly conserved 5′-CAAGUG-3′ sequence was present in motif 2 of KITV. No motifs characteristic of KITV were found in TAKV (100%). In YGTV, motifs 2 and 4 were found in four isolates (66.67%), and motif 4 alone in two isolates (33.33%). In ALSV, motif 2 alone was found in five isolates (62.5%), and no motifs characteristic of KITV were found in the other isolates (37.5%). In HMV, only motif 2 was found in all the isolates (100%). In JMTV, motifs 1, 2, and 4 were present in 32 isolates (59.26%), motifs 1 and 4 in 13 isolates (24.07%), motifs 2 and 1 in 5 isolates (9.26%), motif 1 alone in 2 isolates (3.70%), and 2 isolates (3.70%) lacked any motifs. In MGTV, motifs 1, 2, and 4 were found in all the isolates (100%). The data are shown in Appendix A. On the basis of multiple alignment and the lack of motif 2, we identified the additional 22 JMTV isolates, 2 Yanggou tick virus isolates, and 3 ALSV isolates, which were under-sequenced.

#### 3.7.2. Segment 1’s 3′ UTR

Four conserved motifs were found in segment 1’s 3′ UTR. The discovered motifs are shown in Figure 7a. All analyzed KITV isolates were characterized by motifs 1 and 3. The highly conserved 5′-CAAGUG-3′ sequence was found in motif 4 that is absent in KITV. Motifs 1 and 3 were detected in all the MGTV isolates (100%). Only motif 1 was found in all the TAKV, YGTV, ALSV, and HMV isolates (100%). In JMV, motifs 1 and 3 were detected in 75 isolates (94.9%), and motif 1 alone was present in 4 isolates (5.1%). The data are shown in Appendix A. On the basis of multiple nucleotide sequence alignment and the lack of motif 4, 36 JMV isolates, 2 YGTV isolates, 7 ALSV isolates, 2 MGTV isolates, and 18 KITV isolates were considered to be presumably under-sequenced. The final secondary structure of this region for the fully sequenced KITV 3′ UTR should probably be represented not by a stem loop and a Y-structure but by two Y-structures, as was undertaken for JMTV.

#### 3.7.3. Segment 2’s 5′ UTR

Four conserved motifs were found in segment 2’s 5′ UTR. The discovered motifs are shown in Figure 6b. The highly conserved 5′-CAAGUG-3′ sequence was present in motif 3 of KITV. In isolates KITV/2018/1 and KITV/2018/2, motifs 1, 2, and 3 (11.11%) were found; in isolate KITV/2017/1, in addition to motifs 1, 2, and 3, motif 4 was found (5.56%). Motif 4 arises from the insertion of two adenines at positions 87 and 88 in KITV/2017/1. In KITV isolates (GenBank ID: OP612414.1–OP612428.1), only motifs 1 and 2 (83.33%) were detected. The results of multiple alignment of the 5′ UTRs indicate that they are under-sequenced. The lack of the 5′-GCCGGUGGCAAGUGCAUACAUCGACAACGAAU-3′ region at the beginning of the nucleotide sequence leads to the disappearance of motif 3 that involves the highly conserved 5′-CAAGUG-3′ sequence that is characteristic of the fully sequenced isolates of KITV and other closely related viruses. Motifs 1, 2, and 3 were present in SCWL. In SCTV motifs 1, 2, and 3 were present in two isolates (66.67%), and motifs 1 and 2 in one isolate (33.33%). In MGTV, motifs 1, 2, 3, and 4 were found in one isolate (33.33%). No motifs characteristic of KITV were identified in the other MGTV isolates (66.67%). In ALSV, motif 4 was found in only one isolate (14.29%), and no motifs characteristic of KITV were found in the other isolates (85.71%). No motifs were found in all the YGTV isolates (100%). Only motif 4 was identified in all the HMV isolates (100%). In JMTV, motifs 1, 2, 3, and 4 were present in 35 isolates (43.21%), motifs 2 and 4 in 30 isolates (37.04%), motifs 1, 2, and 4 in 14 isolates (17.28%), no motifs were found in 1 isolate (1.23%), and motifs 1, 2, and 3 were identified in 1 isolate (1.23%). The data are shown in Appendix A. On the basis of multiple alignment and the lack of motif 3, we identified another 44 JMTV isolates, 1 SCTV isolate, 2 MGTV isolates, 6 ALSV isolates, and 2 YGTV isolates, which were presumably under-sequenced.

#### 3.7.4. Segment 2’s 3′ UTR

Five conserved motifs were found in segment 2’s 3′ UTR. The discovered motifs are shown in Figure 7b. Motifs 1, 2, 3, 4, and 5 were characteristic of all the analyzed KITV isolates (100%). The highly conserved sequence 5′-CAAGUG-3′ was present in motif 4. In SCTV, motifs 1, 2, 3, 4, and 5 were present in one isolate (33.33%), motifs 1, 2, 3, and 5 in one isolate (33.33%), and motifs 1, 2, and 3 in one isolate (33.33%). Motifs 1, 2, and 3 were found in SCWL. In MGTV, motifs 1, 2, 3, 4, and 5 were found in one isolate (33.33%), and motifs 1, 2, 3, and 5 were identified in two isolates (66.67%). In ALSV, motif 1 was identified in five isolates (55.56%), motifs 1 and 4 in one isolate (11.11%), and no motifs that were characteristic of KITV in the other isolates (33.33%). In YGTV, motif 4 was found in one isolate (33.33%), and the other isolates (66.67%) lacked motifs characteristic of KITV. All the HMV isolates (100%) had motifs 1 and 4. In JMV, motifs 1, 2, 3, 4, and 5 were present in 44 isolates (53.66%), motifs 1, 2, 3, and 5 in 2 isolates (2.44%), motifs 1, 2, and 3 in 14 isolates (17.07%), motifs 2 and 3 in 8 isolates (9.76%), motifs 1 and 3 in 1 isolate (1.22%), motif 2 in 4 isolates (4.88%), and 9 isolates (10.98%) lacked any motifs. The data are shown in Appendix A. On the basis of multiple alignment and the lack of motif 4, we suggested that 37 JMV isolates, 8 ALSV isolates, 2 YGTV isolates, and 2 MGTV isolates were presumably under-sequenced.

#### 3.7.5. Segment 3’s 5′ UTR

Four conserved motifs were found in segment 3’s 5′ UTR. The discovered motifs are shown in Figure 6c. Motifs 1 and 2 were characteristic of all the analyzed KITV isolates. Motifs 1 and 2, characteristic of KITV, were found in all the MGTV, SCWL, and SCTV isolates (100%). No motifs characteristic of KITV were found in all the HMV, YGTV, and ALSV isolates. In JMTV, motifs 1 and 2 were present in 42 isolates (56.76%), motif 1 alone in 2 isolates (2.70%), and motif 2 alone in 18 isolates (24.32%). The other JMTV isolates (16.22%) lacked motifs characteristic of KITV. The data are shown in Appendix A. On the basis of multiple alignment and the lack of motif 1, we identified another 27 JMTV isolates. Other motifs were found in HMV and ALSV viruses. Interestingly, motif 4 in these viruses also contains the highly conserved 5′-CAAGUG-3′ nucleotide sequence that is typical of motif 1.

#### 3.7.6. Segment 3’s 3′ UTR

Multiple alignment of KITV nucleotide sequences with nucleotide sequences of other closely related viruses revealed the under-sequencing of this region. Further analysis was not performed because similar analysis of segment 1’s 3′ UTR did not provide full information about the conserved motifs and the secondary structure.

#### 3.7.7. Segment 4’s 5′ UTR

Five conserved motifs were found in segment 4’s 5′ UTR. The discovered motifs are shown in Figure 6d. All considered KITV isolates were characterized by the presence of motifs 1, 2, and 3. Motif 3 contained the highly conserved 5′-CAAGUG-3′ sequence. Motifs 1, 2, and 3 characteristic of KITV were found in all the MGTV, SCWL, and SCTV isolates (100%). In all the HMV and ALSV isolates (100%), only motif 3, which is characteristic of KITV, was detected. In YGTV, only one isolate (33.33%) harbored motif 3, whereas two other isolates (66.67%) lacked any motifs characteristic of KITV. In JMTV, motifs 1, 2, and 3 were present in 42 isolates (59.15%), motifs 1 and 2 in 14 isolates (19.72%), motifs 2 and 3 in 1 isolate (1.41%), motif 1 alone in 2 isolates (2.82%), motif 2 alone in 5 isolates (7.04%), motif 3 alone in 3 isolates (4.23%), and the other isolates (5.63%) lacked any motifs characteristic of KITV. The data are shown in Appendix A. On the basis of multiple alignment and the lack of motif 3, we identified another 24 JMTV and 2 YGTV isolates, which were presumably under-sequenced.

#### 3.7.8. Segment 4’s 3′ UTR

Six conserved motifs characteristic of KITV isolates were found in JMV segment 4’s 3′ UTR. The discovered motifs are shown in Figure 7c. Motif 4 comprised the highly conserved 5′-CAAGUG-3′ sequence. In all the MGTV, SCWL, and SCTV isolates (100%), we found motifs 1–6 characteristic of KITV. In all the HMV isolates (100%), only motifs 1 and 4 were detected. In seven ALSV isolates (87.5%), motifs 1, 4, and 6 were present. Only motifs 4 and 6 were found in one ALSV isolate (12.5%). In all the YGTV isolates (100%), only motif 1, which is typical of KITV, was detected. In JMV, all six motifs were present in 51 isolates (53.13%), motifs 1, 2, 3, 5, and 6 were found in 10 isolates (10.42%), motifs 1, 2, 5, and 6 were identified in 24 isolates (25%), motifs 2, 5, and 6 occurred in 3 isolates (3.13%), motifs 1 and 4 were present in only 1 isolate (1.04%), motif 5 alone was detected in 6 isolates (6.25%), and the other isolates (1.04%) lacked motifs characteristic of KITV. The data are shown in Appendix A. On the basis of multiple alignment and the lack of motif 4, we suggested that 42 JMV isolates were presumably under-sequenced.

## 4. Discussion

Orthoflaviviruses are important pathogens that cause serious infections in humans, from mild fever to encephalitis and hemorrhagic fever, which often result in death. JMVs are a recently discovered group of viruses of unknown pathogenicity. They have a segmented RNA genome consisting of four segments, two of which are functionally related to the non-structural genes of viruses from the *Orthoflavivirus* genus, suggesting the existence of segmented flavi-like viruses in the Flaviviridae family.

In this study, we performed whole-genome sequencing of two new KITV/2018/1 and KITV/2018/2 isolates from the collection of *Rhipicephalus* spp. ticks from the city of Kindia, Republic of Guinea. The genomic analysis revealed that the KITV isolates clustered together with the JMTV_1 (MH155892) isolate and the MGTV/V4/11 (JX390986) isolate, which were previously detected in *Rhipicephalus microplus* ticks from the Central-West and Southeast Regions of Brazil and had 90% and 92% nucleotide sequence identity in ORF segment 1 [23]. Interestingly, the KITV isolates from the Republic of Guinea and the JMTV and MGTV isolates from Brazil cluster within a monophyletic clade that is different from that of the isolates identified in Asia, Africa, and Southeast Europe, suggesting that these isolates likely belong to a novel virus species within JMVs.

Comparative analysis of the KITV genome and those of other flaviviruses revealed important structural elements of the 5′ and 3′ UTRs. The 5′ and 3′ UTR of segments 1–3 harbor functional orthoflaviviral 5′ DAR and 3′ DAR regions that are responsible for long-range RNA–RNA interactions during genome cyclization before replication onset [22,24]. The distant location of the 5′ DAR in segment 1 of the KITV genome leads to the emergence of SL-3, SL-4, SL-5, and a Y-structure; additionally, SL-4 arises in segment 2, and the Y-1 structure in segment 3 is modified. When describing a genome cyclization model, the region located downstream of the start codon (5′ DAR) forms hairpin structures, thereby facilitating the replication initiation step. These regions, when they approach each other, activate viral RNA-dependent RNA polymerase [25]. The 5′ UTR of all the segments comprises the 5′-GCAC-3′ sequence that binds the La protein or its complementary 5′-GUGC-3′ sequence, which may facilitate the stabilization of the replication complex [26]. Mutations in the 5′-GCAC-3′ motif are known to alter the binding affinity of La to the hepatitis C virus IRES and have a profound effect on its translation both in vitro and ex vivo [27]. A highly conserved orthoflaviviral 5′-CACAG-3′ pentanucleotide was found in hairpin 3 of the Y-1 structure of the segment 1 3′ UTR. This region is conserved in segment 2’s 3′ UTR only in the KITV/2017/1 isolate and is located in hairpin 7 of Y-3. In flaviviruses, this pentanucleotide is located in the 3′ long stable hairpin (3′ LSH) that is involved in the initiation of genome replication [28]. KITV segment 2’s 3′ UTR is the most heterogeneous. It contains three homologous 5′ UTR sequences R1–R3 and two 40-nucleotide repeats. In the KITV/2018/1 and KITV/2018/2 strains, a 35-nucleotide deletion occurs in the second repeat. In KITV segment 2’s 3′ UTR, two pseudoknots, PK1 and PK2, were found, which are responsible for stabilizing the secondary structure of this region in orthoflaviviruses [29]. These results suggest an unusual evolutionary relationship between the unsegmented and segmented viruses of the Flaviviridae family.

Furthermore, multiple UAG sites (two to six) for interactions with the RNA-binding protein Musashi-1 (MSI1) were found in the 3′ UTR of all the KITV RNA segments. The data are shown in Appendix A. In comparison, African Zika virus and European tick-borne encephalitis virus contain six MSI1 binding sites. In this case, the saturation coefficient of the 3′ UTR with the UAG trinucleotides in Zika virus is 1.09, which indicates a high level of binding sites relative to the 3′ UTR’s size, whereas the same coefficient in tick-borne encephalitis virus is 0.55 [30]. According to in vivo studies, MSII enhances virus replication, regulates translation, and is involved in Zika virus-induced neurotropism through interaction with the 3′ UTR of Zika virus RNA [31,32]. Zika virus RNA may compete with endogenous targets for the binding of MSI1 in the developing embryonic brain, thereby dysregulating the expression of genes essential for neural stem cell development, which suggests the presence of mechanisms of MSI1-mediated congenital neuropathology [33]. Therefore, many binding sites of the 3′ UTR of KITV to MSI1 may indicate a possible neurotropic potential of this virus, which requires further research to clarify the epidemiological significance of this unusual multicomponent flavi-like virus.

The 5′ UTR and 3′ UTR of segmented flavi-like viruses have been found to consist of patterns of repeating sequences (conserved motifs). The structure and number of conserved motifs are individual to each virus. However, the 5′ UTR and 3′ UTR of all the segments contained the 5′-CAAGUG-3′ sequence in at least one conserved motif. These data are consistent with a previously published paper on the search for conserved sequences in the untranslated regions of JMVs [12]. Analysis of the obtained models revealed that the 5′-CAAGUG-3′ sequence often forms the first hairpin of the 5′ UTR and hairpins of the 3′ UTR Y-1 structure. These regions are shown in Appendix A, respectively. KITV is also characterized by the presence of 5′-CAAGUG-3′ in hairpin 1 of the 5′ UTR of segments 1, 2, and 4 and in the hairpins of the Y-1 structure of the 3′ UTR of segments 2 and 4. As reported, JMTV retained the 5′-CAAGUG-3′ sequence in the UTR of all the segments during its first isolation, which indicated that all four segments belonged to the same virus [2]. The conserved location of the 5′-CAAGUG-3′ sequence in the first structures of the untranslated regions may indicate that this region is involved in interactions with viral and/or cellular proteins to regulate the virus’s life cycle, similarly to how it occurs in other viruses of the *Orthoflavivirus* genus [34,35,36,37,38].

## 5. Conclusions

In this study, we obtained two complete genome sequences and modeled, for the first time, the 5′ and 3′ UTRs of the recently discovered flavi-like Kindia tick virus with a segmented genome. The regulatory elements identified during the analysis of the 5′ and 3′ UTR structures indicate an evolutionary link between JMVs and classical orthoflaviviruses. Further investigation into each individual component of the main 5′ and 3′ UTR elements will promote an understanding of routes for implementation of genetic information in novel multicomponent flavi-like viruses. The results of this study may be helpful in future research aimed at investigating the epidemiology, structural organization, replication, and pathogenesis of these mysterious multicomponent flavi-like viruses of the JMV group.

## Figures and Tables

**Figure 1 microorganisms-12-00239-f001:**
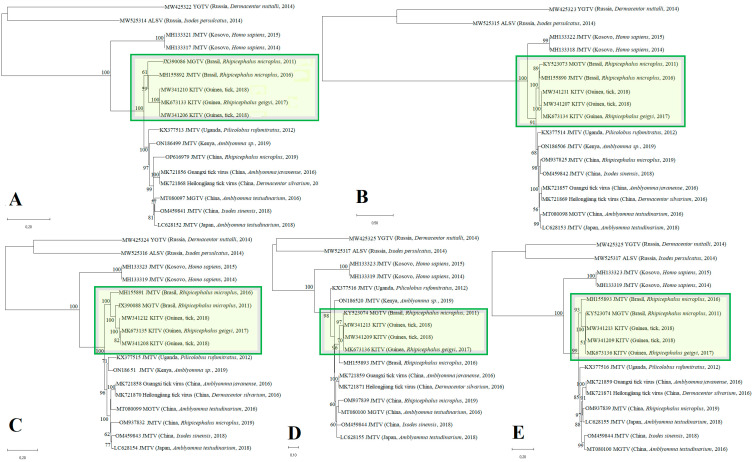
Phylogenetic analysis of tick-borne flavi-like viruses with a segmented genome according to the nucleotide sequence ORF of the all segments. (**A**) Phylogenetic analysis of the ORF nucleotide sequence encoding RNA-dependent RNA polymerase. (**B**) Phylogenetic analysis of the ORF nucleotide sequence encoding VP1. (**C**) Phylogenetic analysis of the ORF nucleotide sequence encoding the NS3 viral protein. (**D**) Phylogenetic analysis of the ORF nucleotide sequence encoding VP2. (**E**) Phylogenetic analysis of the ORF nucleotide sequence encoding VP3.

**Figure 2 microorganisms-12-00239-f002:**
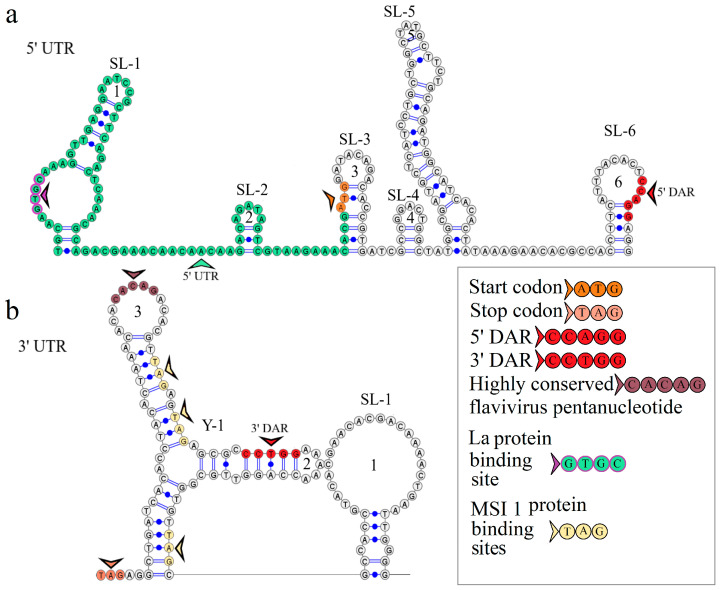
Linear models of the secondary structure of 5′ UTR (**a**) and 3′ UTR (**b**) of genomic RNA segment 1 of the KITV isolate KITV/2018/1. Functional orthoflavivirus regions are indicated by colored arrows.

**Figure 3 microorganisms-12-00239-f003:**
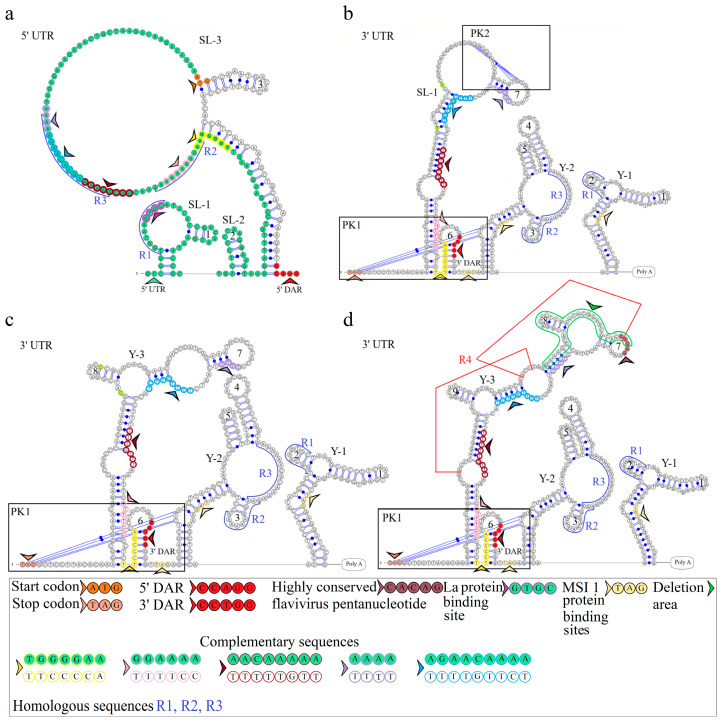
Linear models of the secondary structure of 5′ UTR and 3′ UTR of genomic RNA segment 2 of KITV (**a**). 5′ UTR of the KITV/2018/1 isolate (**b**). 3′ UTR of the KITV/2018/1 isolate. (**c**). 3′ UTR of the KITV/2018/2 isolate (**d**). 3′ UTR of the KITV/2017/1 isolate. The complementary sequences in 5′ and 3′ UTR are highlighted with colored circles. Homologous sequences in 5′ and 3′ UTR are designated as R1, R2, R3.

**Figure 4 microorganisms-12-00239-f004:**
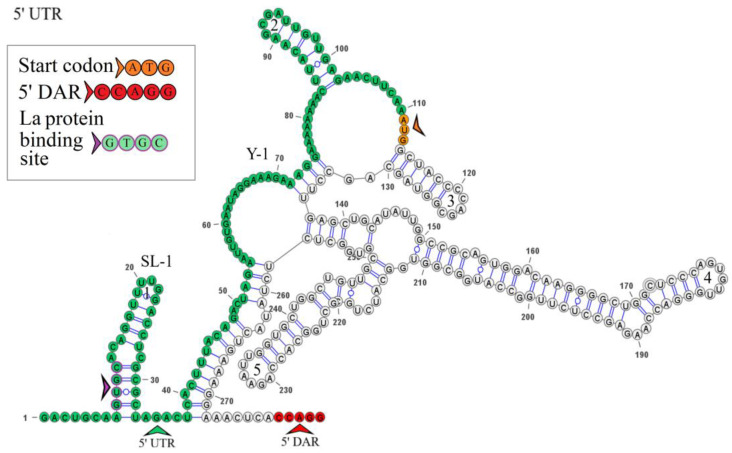
Linear model of the secondary structure of the 5′ UTR of genomic RNA segment 3 of the KITV/2018/1 isolate. Functional orthoflavivirus regions are indicated by colored arrows.

**Figure 5 microorganisms-12-00239-f005:**
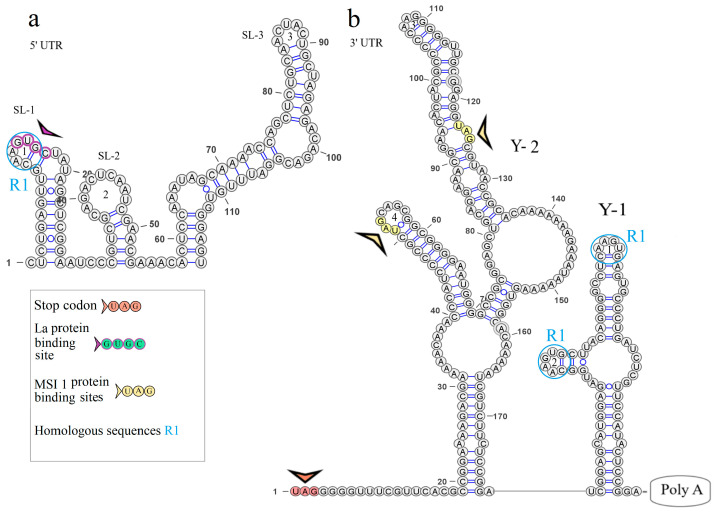
Linear models of the secondary structure of 5′ UTR (**a**) and 3′ UTR (**b**) of genomic RNA segment 4 of the KITV isolate KITV/2018/1. Functional orthoflavivirus regions are indicated by colored arrows.

**Figure 6 microorganisms-12-00239-f006:**
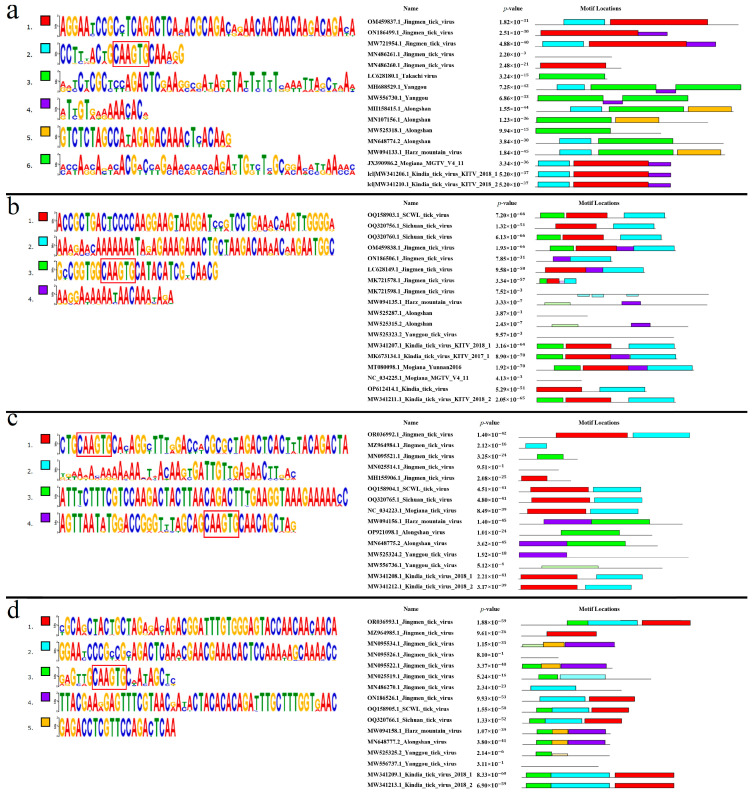
Conservative motifs in 5′ UTR segmented flavi-like viruses. (**a**). Nucleotide sequences of motifs and their location in the 5′ UTR of segment 1 of segmented flavi-like viruses. A highly conservative area is highlighted in motif 2. (**b**). Nucleotide sequences of motifs and their location in the 5′ UTR of segment 2 of segmented flavi-like viruses. A highly conservative area is highlighted in motif 3. (**c**). Nucleotide sequences of motifs and their location in the 5′ UTR of segment 3 of segmented flavi-like viruses. A highly conservative area is highlighted in motifs 1 and 4. (**d**). Nucleotide sequences of motifs and their location in the 5′ UTR of segment 4 of segmented flavi-like viruses. A highly conservative area is highlighted in motif 3. The size of the bases correlates with the conservation of their detection among different JMV isolates. Complete date are presented in Appendix A.

**Figure 7 microorganisms-12-00239-f007:**
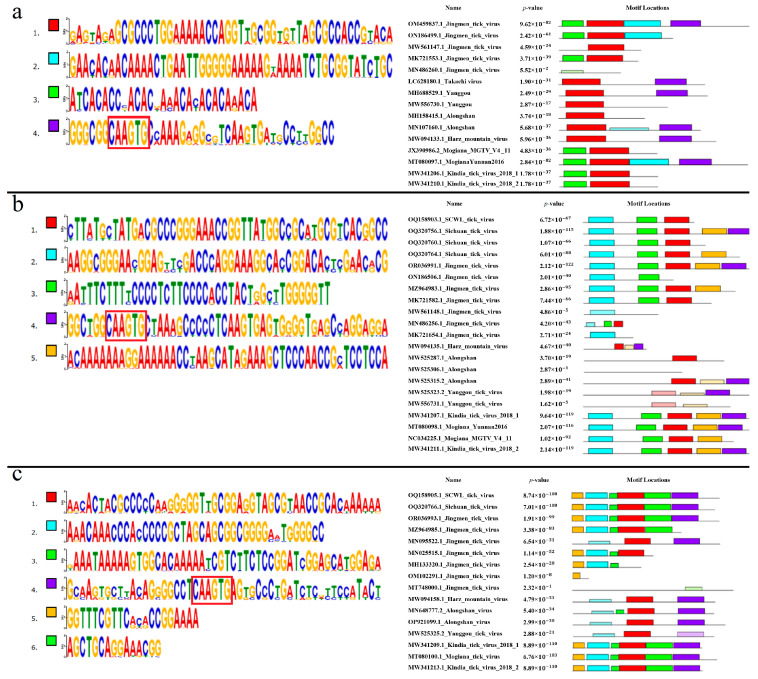
Conservative motifs in 3′ UTR segmented flavi-like viruses. (**a**). Nucleotide sequences of motifs and their location in the 3′ UTR of segment 1 of segmented flavi-like viruses. A highly conservative area is highlighted in motif 4. (**b**). Nucleotide sequences of motifs and their location in the 3′ UTR of segment 2 of segmented flavi-like viruses. A highly conservative area is highlighted in motif 4. (**c**). Nucleotide sequences of motifs and their location in the 3′ UTR of segment 4 of segmented flavi-like viruses. A highly conservative area is highlighted in motif 4. The size of the bases correlates with the conservation of their detection among different JMV isolates. Complete date are presented in the Appendix A.

**Table 1 microorganisms-12-00239-t001:** Comparative sizes of 5′ and 3′ UTRs of KITV and MGTV.

Name of the Isolate of KITV and MGTV, Place of Isolation	Size of UTRs							
	Segment 1		Segment 2		Segment 3		Segment 4	
	5′ UTR	3′ UTR	5′ UTR	3′ UTR	5′ UTR	3′ UTR	5′ UTR	3′ UTR
KITV/2018/1, Guinea, Africa	91	127 *	154	352	110	177 *	130	247
KITV/2018/2, Guinea, Africa	91	127 *	154	352	100	143 *	130	247
KITV/2017/1, Guinea, Africa	104	119 *	156	387	97	143 *	130	244
MGTV/V4/11, Brazil, South America	91	130 *	51 *	318 *	102	179 *	130	250
MGTV Yunnan2016, China, Asia	105	226	176	348	117	254	135	258
Other KITV, Guinea, Africa	OP612399.1–OP612413.1		OP612414.1–OP612428.1		OP612443.1–OP612439.1		OP612458.1–OP612444.1	
	91	127 *	122 *	352	99	175 *	130	247
*Orthoflavivirus*								
Name of the virus			Size of 5′ UTR			Size of 3′ UTR		
Zika virus			107			428		
Tick-borne encephalitis virus			132			428–764		

Note: *—the region is completely under-sequenced.

## Data Availability

All obtained sequences were deposited in GenBank.

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
