# Peer review of "Untranslated Regions of a Segmented Kindia Tick Virus Genome Are Highly Conserved and Contain Multiple Regulatory Elements for Viral Replication"

_microorganisms, 2024, doi:10.3390/microorganisms12020239_

Round 1

Reviewer 1 Report

Comments and Suggestions for Authors

1. Abbreviations that first appear without corresponding full names or other similar errors:

Line 40: SCWL et al.;

Some abbreviations are repeated, such as JMVS in Line 33;

The abbreviation and full name of Line 439's LSH are reversed.

2. Writing errors in bold and italics:

The Rhipicephalus of Line 141 requires italics;

The Flaviviridae of Line 411 do not require italics.

3. Other writing issues:

Perhaps replacing the possible in Line 43 with potential would be better;

Line 46: 4 (5) should be written as 4 (or 5).

4. Image table issues:

The resolution of Figure 1 in Line 161 is too low.

The Note section in the title of Table 1 of Line 177 should be placed below the table. There is an issue with the information in the second and third rows of the table header, resulting in duplicate content.

Figure 4 of Line 253 lacks annotation information within the image.

The suffix names for different segments of KITV/2018/1 in Figure 6abcd of Line 386 and Figure 7abc of Line 397 are not consistent. Some are written as Kladia tick virus KITV/2018/1, some as KITV/2018/1, and some as Kladia tick virus-2018-1. Also, there is no information about KITV/2018/2 in the picture. Should be done even if it is exactly the same as KITV/2018/1?

5. Questions that exist in the results section:

There are some issues with the 3.1 Phylogenetic tree and substitution rate section of Line 139. Firstly, the similarity between KITV/2018/1 and KITV/2018/2 and KITV/2017/1 is very high. Why is it considered that KITV/2018/1 and KITV/2018/2 are new viruses? More evidence is needed to confirm this part of the results. Secondly, the article only used segment1 to construct an evolutionary tree, which requires clear evidence that segment1 alone can represent the entire virus, or to supplement the evolutionary tree constructed by segment2, segment2, segment3, segment4, and 5 segments.

The 3.5 KITV segment 3 5 'and 3' UTR structures section of Line 241 does not provide results for segment 3 3 'UTR structures, perhaps the title needs to be modified.

Author Response

Thank you for reviewing our manuscript (microorganisms-2832257) entitled “UNTRANSLATED REGIONS OF A SEGMENTED KINDIA TICK VIRUS GENOME ARE HIGHLY CONSERVED AND CONTAIN MULTIPLE REGULATORY ELEMENTS FOR VIRAL REPLICATION” submitted for publication in Microorganisms. 

We thank you for valuable suggestions that allowed us to make the manuscript more convincing and understandable. We accepted your suggestion and made corresponding change in the manuscript. Also, we carried out additional construction of the evolutionary tree for segment 2, segment 3, segment 4 of Kindia tick virus as you recommended.

Below please find our detailed responses to your questions and comments. All modifications in the manuscript have been highlighted in red. Please see the attachment.

Reviewer 2 Report

Comments and Suggestions for Authors

In the article entitled: “Untranslated regions of a segmented Kindia tick virus genome are high conserved and contain multiple regulatory elements for viral replication”, the authors obtained whole genome sequences of two Kindia tick virus (KITV) isolates and performed modeling and functional annotation of the secondary structure of 5’ and 3’ UTRs from JMV and KITV viruses. 

Introduction: what is the medical concern of Jingmenviruses and the relative meaning of doing modeling and functional annotation of the secondary structure?

Line 69-72: where does the two isolated originate respectively?

Line 117: What is the justification for selecting 35 nucleotides as the threshold for the selection criteria?

Line 157-159: The sentence appears to be unclear or confusing.

Table 1: Why did the authors choose to compare KITV and MGTV instead of other viruses? Additionally, how do the undersequenced regions impact the results?

Figure6-Figure7: please explain the reasons for the differing size of the bases.

Comments on the Quality of English Language

nope

Author Response

Thank you for reviewing our manuscript (microorganisms-2832257) entitled “UNTRANSLATED REGIONS OF A SEGMENTED KINDIA TICK VIRUS GENOME ARE HIGHLY CONSERVED AND CONTAIN MULTIPLE REGULATORY ELEMENTS FOR VIRAL REPLICATION” submitted for publication in Microorganisms. 

We thank you for valuable suggestions that allowed us to make the manuscript more convincing and understandable. We accepted your suggestion and made corresponding change in the manuscript. Below please find our detailed responses to your questions and comments. All modifications in the manuscript have been highlighted in red. Please see the attachment.
